# Microfluidic Fabrication of Gadolinium-Doped Hydroxyapatite for Theragnostic Applications

**DOI:** 10.3390/nano13030501

**Published:** 2023-01-26

**Authors:** Manuel Somoza, Ramón Rial, Zhen Liu, Iago F. Llovo, Rui L. Reis, Jesús Mosqueira, Juan M. Ruso

**Affiliations:** 1Soft Matter and Molecular Biophysics Group, Department of Applied Physics, University of Santiago de Compostela, 15782 Santiago de Compostela, Spain; 23B’s Research Group, I3Bs—Research Institute on Biomaterials, Biodegradables and Biomimetics, University of Minho, Headquarters of the European Institute of Excellence on Tissue Engineering and Regenerative Medicine AvePark—Parque de Ciência e Tecnologia Zona Industrial da Gandra Barco, 4805-017 Guimarães, Portugal; 3ICVS/3B’s—PT Government Associate Laboratory, 4806-909 Braga, Portugal; 4Department of Physics and Engineering, Frostburg State University, Frostburg, MD 21532, USA; 5QMatterPhotonics, Departamento de Física de Partículas, Universidade de Santiago de Compostela, 15782 Santiago de Compostela, Spain; 6Institute of Materials (iMATUS), Department of Applied Physics, Universidade de Santiago de Compostela, 15706 Santiago de Compostela, Spain

**Keywords:** Computational Fluid Dynamics (CFD), nanomaterials, microfluidics, tissue engineering, theragnostic

## Abstract

Among the several possible uses of nanoparticulated systems in biomedicine, their potential as theragnostic agents has received significant interest in recent times. In this work, we have taken advantage of the medical applications of Gadolinium as a contrast agent with the versatility and huge array of possibilities that microfluidics can help to create doped Hydroxyapatite nanoparticles with magnetic properties in an efficient and functional way. First, with the help of Computational Fluid Dynamics (CFD), we performed a complete and precise study of all the elements and phases of our device to guarantee that our microfluidic system worked in the laminar regime and was not affected by the presence of nanoparticles through the flow requisite that is essential to guarantee homogeneous diffusion between the elements or phases in play. Then the obtained biomaterials were physiochemically characterized by means of XRD, FE-SEM, EDX, confocal Raman microscopy, and FT-IR, confirming the successful incorporation of the lanthanide element Gadolinium in part of the Ca (II) binding sites. Finally, the magnetic characterization confirmed the paramagnetic behaviour of the nanoparticles, demonstrating that, with a simple and automatized system, it is possible to obtain advanced nanomaterials that can offer a promising and innovative solution in theragnostic applications.

## 1. Introduction

Through interdisciplinarity, the intense research activity dedicated to nanotechnologies led to the development of multifunctional nanoparticles (NPs) adapted to various applications in fields such as energy, electronics, catalysis, and medicine [1,2,3,4,5,6,7]. Specifically in the latter, NPs show interesting properties: a high surface/volume ratio that induces a different chemical reactivity than bulk materials, the possibility of gathering considerable amounts of therapeutic agents, and also finely adapting their size, shape, and surface to modulate their biodistribution. Among the different medical applications, theragnostic is one of the most appealing. This field has focused particularly on magnetic nanoparticles (MNPs) [8] and its main imaging modality is magnetic resonance imaging (MRI). Likewise, its functionality is greater, as it has the ability to be combined with other techniques such as computed tomography (CT) [9], fluorescence imaging [10], proton emission tomography (PET) [11], or single photon emission computed tomography (SPECT) [12] for multimodal images. On the therapeutic side, MNPs are being studied for drug delivery [13], phototherapy [14], photodynamic therapy [15], and radiotherapy [16]. In particular, radiotherapy is of particular interest in the context of theragnostic applications because it is the most widely used non-surgical therapy [17] and will greatly benefit from the contribution of imaging.

Current design strategies are devoted to bringing together the same magnetic platform PET/SPECT imaging in combination with MRI for the individualization of radiotherapy treatment, its evaluation, and follow-up [18]. For such treatments, the development of compounds that can be used effectively as imaging agents and radiosensitizers (i.e., compounds that can enhance the effect of radiotherapy) could be a real breakthrough. Most of the radiosensitizers tested are composed of elements with a high atomic number Z that show significantly higher mass energy absorption properties compared to soft tissues [19]. Some classical molecular contrast agents based on iodine, gadolinium, and some molecular anticancer drugs have already been tested for radiosensitization [20]. Due to their ability to contain a larger number of high-Z elements, radiosensitizing NPs received much attention. Among them, Gadolinium (Z = 64) seems very attractive for the design of NPs, as it behaves as a radiosensitizer as well as an MRI contrast agent, conditions that make Gd one of the best-known lanthanide elements for medical applications. Its most common oxidation state is Gd^3+^ similar to other lanthanides. Furthermore, it is considered a hard acid and shows large coordination numbers (8–10) [21]. Its use in medicine is directly related to the development of MRI and contrast agents for this imaging modality. In this way, a large number of studies evaluating the use of Gd-based nanoplatforms for MRI purposes have been reported [22,23]. Its choice is based on the seven unpaired electrons (mostly stable paramagnetic ions) found in the 4f shell of Gd ions, its slow electronic relaxation [24], and the high paramagnetic properties well suited for biomedical applications [25]. In imaging, it improves MRI images by decreasing the T1 relaxation constant of the tissues in which it was accumulated [26,27]. In other words, under T1-enhanced images, Gd reduces the longitudinal relaxation time of related water protons in its coordination sphere creating a bright contrast. Briefly, when the Gd compound is accessible, the solvent (water) related relaxation rate would increase due to dipole-dipole interactions between the nuclear spin of the proton and the fluctuating local magnetic field resulting from the unpaired electron spins [28]. In addition to their use for imaging applications, Gd-based compounds are also envisioned for therapeutic applications. In fact, Gd shows a relatively high atomic number and, therefore, can interact with many types of radiation (X or gamma rays, neutrons, electrons, and hadrons, among others) [29]. Although this interaction can be used for TC [30], its real interest lies in radiosensitization as molecular compounds. Among the different lanthanides, Gd is also of particular interest for neutron capture therapy due to the high neutron capture cross-section of non-radioactive 157Gd [31].

For its part, Hydroxylapatite (HAp) is a calcium phosphate with the formula Ca_10_(PO_4_)_6_(OH)_2_ and is the main inorganic component of vertebrate bone. In its synthetic form, it is the most widely used bioceramic for the repair and regeneration of bone tissue defects, since it has characteristics such as biocompatibility, non-toxicity, chemical stability, osteoconduction, and bioactivity. Recent advances in the preparation of nano-sized HAp with customized surface characteristics and colloidal stability in combination with mentioned properties have opened new perspectives in their use in non-bone related applications [32,33,34]. HAp has a highly flexible structure that can easily incorporate foreign ions, inducing changes in its physical-chemical properties. It was observed that HAp NPs can be endowed with a range of abilities that mere HAp does not display; as photocatalytic, luminescent, and magnetic properties through different doping [35,36,37]. In this sense, the use of Gd^3+^ doped HAp NPs has been proposed for use as multifunctional or theragnostic nanomaterials that are currently at the forefront of nanomedicine. The reason for the doping of HAp with Gd is due to the fact that the similar ionic radii between Gd(III) and calcium Ca(II) can give rise to a possible substitution of Ca by Gd in some Ca(II) binding sites. We have previously dealt with hydroxyapatite nanoparticles on multiple occasions. We demonstrated that the presence of pre-embedded HAp nanoparticles on gelatin scaffolds has a clear influence on its mechanical performance and its mineralized phase deposition epitaxial development [38,39]. We developed periosteum-inspired bilayered membranes [40] and also investigated HAp nanoparticle interactions with globular proteins [41]. For these works, we used a technique formerly proposed by Liu et al. [16] to synthesize HAp nanoparticles. Nevertheless, in order to improve and optimize the process we created a new route for its synthesis using microfluidic devices [42]. This approach is based on a network of microchannels that are coupled to each other within a “microfluidic chip” to perform a specific purpose. Inlets and outlets, perforated on the chip, link the channels to the outside, where liquids are introduced by syringe pumps or pressure gauges at a constant flow rate or constant pressure, respectively. These systems make use of two key characteristics: compact size and laminar flow, which allows for greater process control [43]. These benefits, together with minimal reagent use, superior resolution, and quicker analytical times, explain the tremendous growth of this field in recent years [44,45]. Microfluidics systems, in particular, have shown a remarkable potential for modulating important phases in nanosynthesis, including nucleation, growth, and reaction conditions, in order to efficiently tune the size, polydispersity, shape, and repeatability of nanoparticles [46]. Furthermore, unlike traditional production systems, having continuous flow allows for online monitoring, which leads to improved control. Apart from that, the systems’ modest length scales promote heat/mass transport and produce mixing efficiency [47]. Fluid flows within the microchannels at a regulated rate when injected at constant flow rates. However, at these scales, it is not easy to monitor the nucleation and developmental stages in the creation of nanoparticles as a function of distance from the injection site. Computational methods, primarily those based on finite elements, have made it possible to determine many of these processes. However, the vast majority of these studies focus on the flows inside the microchip, ignoring what happens in both the inlet and outlet tubes of the device [48].

In this work, we take advantage of the medical potential applications of Gadolinium as a contrast agent with the versatility and huge array of possibilities that microfluidics can provide. Furthermore, in order to have global control of the process from the first instant to the last, we have performed simulations covering all phases and all elements of our device. As a result, particle size, distribution, and morphology can be precisely controlled. The result is a route and methodology capable of efficiently and functionally creating advanced nanomaterials with precise structural and magnetic properties that can offer a promising and innovative solution in theragnostic applications.

## 2. Materials and Methods

### 2.1. Reagents

Poly (propylene glycol) (PPG, MW = 425 g·mol^−1^, Sigma, Madrid, Spain), hexadecyl-trimethyl ammonium bromide (CTAB, MW = 364.48 g·mol^−1^, 99%, Sigma), sodium phosphate (Na_3_PO_4_, MW = 148 g·mol^−1^, 96%, Sigma), sodium nitrite (NaNO_2_, MW = 69 g·mol^−1^, 97%, Sigma), calcium chloride (CaCl_2_, MW = 91 g·mol^−1^, 99%, Sigma), gadolinium oxide (Gd_2_O_3_, MW = 362.50 g mol^−1^, 99.9%, Sigma) hydrochloric acid (HCl, MW = 36.46 g·mol^−1^, Sigma), and Poly(dimethylsiloxane) vinyl (PDMS) ((C_2_H_6_OSi)_n_, Sylgard 184, Dow Corning, Midland, Michigan, United States) were used for the synthesis directly without purification. Triple distilled water was used to prepare all solutions.

### 2.2. Thermal Dehydration of GdCl_3_·6H_2_O Crystalline Hydrate

Pure gadolinium oxide, Gd_2_O_3_, was put into a glassy carbon cup and dissolved in an excess of concentrated HCl to produce crystalline hydrate GdCl_3_·6H_2_O, by slowly heating the solution to temperatures no greater than 100 °C [49]:Gd_2_O_3_ + HCl_conc. (exc.)_ → GdCl_3_⋅6H_2_O + H_2_O↑(1)

The crystalline hydrate of gadolinium trichloride GdCl_3_·6H_2_O was subsequently dehydrated by heating the sample in the temperature range of 25–200 °C, under the following sequence of chemical reactions:GdCl_3_⋅6H_2_O → GdCl_3_⋅3H_2_O + 3H_2_O↑,(2)
GdCl_3_⋅3H_2_O → GdCl_3_⋅2H_2_O + H_2_O↑
GdCl_3_⋅2H_2_O → GdCl_3_ + H_2_O↑,

Then, the hydrolysis continues in the temperature range of 200–275 °C, helping the anhydrous salt along with the gadolinium oxychloride product:GdCl_3_⋅H_2_O → GdCl_3_ + H_2_O↑.(3)
GdCl_3_⋅H_2_O → GdOCl + HCl↑.

Consequentially, the complete sequence of the thermal dehydration of gadolinium chloride crystalline hydrate can be summarized as follows:Gd_2_O_3_ + HCl_conc. (exc.)_ → 2GdCl_3_⋅6H_2_O → 2GdOCl + 2HCl↑ + 5H_2_O↑.(4)

Being the final product a mixture of gadolinium chloride anhydrous with the presence of a phase of gadolinium oxychloride. The formation of the GdOCl phase was confirmed by the XRD. This powder is then blended with calcium chloride at different ratios. As a reagent in the synthesis, said combination is employed to achieve partial substitution of the calcium ion with paramagnetic ions of gadolinium. As a consequence, doped nanoparticles with magnetic properties for theragnostic usage are produced: Ca_10−x_Gd_x_(PO_4_)_6_(OH)_2_.

### 2.3. Fabrication of the Microchips

The lithography technique was used to build microfluidic devices. The mould was created by exposing a 2” diameter silicon wafer to oxygen plasma to polish its surface before spin coating it with an epoxy resin at 500 rpm for 75 s to control the size of the channels (500 µm). A prebaking treatment in a hot plate at 60 °C for 8 min and 95 °C for 15 min eliminated the resin’s solvent. After the resin had hardened on the substrate, a mask was applied, and the resin was cross-linked for 4 min using a UV light (365 nm). The substrate was then immersed in propylene glycol monomethyl ether acetate (PGMEA) for 10 min to remove the substrate’s nonexposed resin. The mould was then baked at 135 °C for 2 h to improve its robustness. Following that, polydimethylsiloxane (PDMS) was placed into the created mould and baked for 45 min at 60 °C until complete solidification was achieved. After obtaining the PDMS chips, the final step was to attach them to the surface of the microscope slides. To this end, the slides’ surfaces were additionally coated with PDMS and dried. The chips and slides were then placed in a plasma cleaner for 1 min to activate their surfaces. Once in contact, the microscope slides and chips were correctly fused and consolidated, completing the procedure and yielding the final microchips.

### 2.4. Computational Parameterization

Lab-on-a-chip (LOCs) follow the laws of fluidics, specifically those of microfluidics, giving rise to some particularities and approximations caused by the small dimensions with which they work. There are many parameters and laws that enter this field, but the most relevant elements are the Navier-Stokes Equation (5) and the Reynolds Number Equation (6). The first expression shows the behaviour of the liquid within the specified geometry. For its part in solving the second expression, it is possible to know the conditions under which the system is working, meaning that if the Reynolds number is less than 2000 units, the flow is laminar. This requisite is necessary to guarantee homogeneous diffusion between the elements or phases in play. Additionally, the approximation of assuming that the fluid is a continuous medium and not a set of particles, of which mention is made with the Knudsen number [50]. Specifically, the parameters used in the LOCs are adjusted to the continuous model.

(5)
$$\rho \frac{D\phi }{Dt}-\nabla \cdot \left(-pI+\mu (\nabla \phi +{\left(\nabla \phi \right)}^{T}\right)-Fv=0$$


(6)
$$Re=\frac{\rho \;U^{2}}{\mu \frac{U}{d}}=\frac{\rho \;d\;U}{\mu }$$


In Equation (5), 
ϕ
 is the flow, 
ρ
 represents the density of the medium, 
p
 the pressure, 
μ
 the viscosity of the medium, 
I
 symbolizes the identity tensor, and *Fv*, the applied volume forces. Referring to Equation (6), 
d
 is the characteristic length and 
U
 the fluid velocity. Furthermore, it is shown that if the Reynolds number is introduced in the expression of the Navier-Stokes equation for the description of a laminar flow, it is determined that if the parameter is small, the term can be neglected, which eliminates the parameter of the inertial effects of the flow [51]:
(7)
ReDϕ′Dt′−∇′·−p′I+(∇′ϕ′+∇′ϕ′T−Fv=0


### 2.5. Microfluidic Procedure for the Doped—HAp Synthesis

A microfluidic system comprised of two coupled Y-junction chips was used to create the HAp nanorods [42] (Appendix A). The initial concentrations of each stock were recalculated by setting the input ratios of each syringe to achieve the same concentrations as in the traditional procedure [52] (Appendix A). The microfluidic synthesis process is composed of three inputs. Stock A_i_, (i.e., the calcium precursor alone, or together with the corresponding concentrations of Gd, ratio Ca/Gd, 1:1 and 1:10, respectively), is introduced into one of the syringes. Through the remaining input of the first microchip, the micellar solution and sodium nitrite (stock B) are injected. This mixture flows through the channel of the first microreactor resulting in the formation of the CTAB/PPG micellar template that give the rods their final shape. This mixture is then connected to the second microchip, and, at that point, the phosphate (sock C) is also added at the same flow rate. In this process, CTAB interacts with the phosphate groups, attracting calcium ions (or gadolinium) and inducing the nucleation of HAp crystals at the micelle interface.

The inputs were linked to syringe pumps (KDS 101 Legacy Syringe Pump) which were calibrated to provide a precise and steady flow in each of the three inlets. After the synthesis, the resulting mixture was purified using heat treatment with the aim of separating the HAp nanorods from the salts and organic compounds of the micellar solution. Afterward, to remove impurities, it was autoclaved at 100 °C for 24 h before being filtered and rinsed with distilled water. The material was then dried for 24 h at 50 °C before being finally calcinated for 3 h at 400 °C in the muffle furnace. After the complete procedure, white HAp powders with different doping rates were obtained: pure HAp, HAp with a 1:1 (Ca/Gd) doping ratio, and HAp with a 1:10 (Ca/Gd) doping ratio. The samples are referred to as follows: HAp, HAp:Gd1, and HAp:Gd10, respectively. Additionally, a fourth sample using Mg as a doping agent (1:1 ratio (Ca/Mg)) was used as a control (referred to as HAp:Mg).

### 2.6. Physicochemical Characterization

#### 2.6.1. FE-SEM with EDX

A field emission scanning electron microscope (FE-SEM ZEISS UTRA PLUS) was used to examine surface morphology. A Secondary Electron Detector (in the lens) was utilized to collect all the SEM images. The FE-SEM is equipped Energy Dispersive X-Ray (EDX) spectroscopy system. The accelerating voltage (EHT) was 3.00 kV, with a resolution (WD) of 2.1 nm. Finally, for the studied samples it was not necessary to compensate for the charge locally or to shade the sample by introducing nitrogen gas.

#### 2.6.2. Quantitative Evaluation of the Surface Roughness and Topography

To evaluate the surface profiles of the samples, the surface roughness was analyzed. Using digitized field emission scanning electron microscopy (FE-SEM) images and ImageJ software (v. 1.53t, August 2022, National Institutes of Health, Bethesda, MD, USA), the arithmetic mean roughness deviation *R_a_*, root mean square roughness *R_q_*, kurtosis *R_ku_*, and skewness coefficient *R_sk_* were calculated. The parameters are briefly described below.

The arithmetic average height parameter *R_a_* is the most widely used roughness parameter for general quality control. It is defined as the average absolute deviation of the roughness irregularities from the mean line over the sampling length. The mathematical definition and digital implementation of the arithmetic mean height parameter are:

(8)
$$R_{a}=\frac{1}{n}\int_{0}^{l}\left|y\left(x\right)\right|dx\;R_{a}=\frac{1}{n}\overset{n}{\underset{i=1}{\sum }}\left|y_{i}\right|$$


To obtain the standard deviation of the surface height distribution, the root mean square of the roughness *R_q_* is used. It is an important parameter to describe the surface roughness by statistics. This parameter is more sensitive than *R_a_* to a large deviation from the midline. The mathematical definition and digital implementation are given by:
(9)
$$R_{q}=\sqrt{\frac{1}{l}\int_{0}^{l}{\left\{y\left(x\right)\right\}}^{2}dx}\;R_{q}=\sqrt{\frac{1}{n}\overset{n}{\underset{i=0}{\sum }}y_{i}^{2}}$$


Next, the asymmetry of the profile was analyzed to measure the symmetry of the roughness profile with respect to the midline. This parameter is sensitive to a longitudinal imbalance between valleys and peaks. A symmetric height distribution, that is, with peaks identical to the valleys, has zero skewness. The skewness parameter *R_sk_* can be used to distinguish between two profiles that have the same *R_a_* or *R_q_* values but different shapes. The mathematical and numerical formula used to calculate the asymmetry of a profile, which has a number *N* of points and *Y_i_* being the height of the profile at point i, is as follows:

(10)
$$R_{sk}=\frac{1}{R_{q}^{3}}\int_{-\infty }^{\infty }y^{3}p\left(y\right)dy\;R_{sk}=\frac{1}{NR_{q}^{3}}\left(\overset{N}{\underset{i=1}{\sum }}Y_{i}^{3}\right)$$


The kurtosis coefficient *R_ku_* describes the sharpness of the profile probability density. If *R_ku_* < 3, the curve is said to have relatively few high peaks and low troughs and follows a smooth sinusoidal trend. If *R_ku_* > 3, the distribution curve is said to have several high peaks and deep valleys. Its mathematical definition is:

(11)
$$R_{ku}=\frac{1}{R_{q}^{4}}\int_{-\infty }^{\infty }y^{4}p\left(y\right)dy\;R_{ku}=\frac{1}{NR_{q}^{4}}\left(\overset{N}{\underset{i=1}{\sum }}Y_{i}^{4}\right)$$


In addition, fractal architecture was evaluated, as this concept is particularly interesting in surface and material science. Natural fractals are repeating patterns on a finite range of scales, and many biological structures are fractals. To calculate the fractal dimension *D_f_*, the box count method is applied to different FE-SEM images. This protocol has been used previously [53] and consists of applying an increasingly finer grid over the studied area and counting, in each interaction, the number of boxes that contain at least part of the object to be measured. The fractal dimension *D_f_* is then linked to the number *n*(s) of boxes of dimension s needed to fill the surface area of the particle according to Ref. [54]:
(12)
$$D_{f}=\underset{s\to 0}{\lim}\frac{\ln\;n\left(s\right)}{\ln\;\left(1/s\right)}$$


This method was optimized by a calculation procedure that allows a precise determination of the key parameters: the number and dimensions of the boxes. The images, initially 256 grey levels and 1024 × 768 pixels in size, are converted to binary images. The fractal dimension is then derived from the slope of a linear least squares fit of the plot of ln (n) vs. ln (box size), where n is the number of equal, non-overlapping boxes that would fill the projected surface area of the aggregate [55,56].

#### 2.6.3. X-ray Powder Diffraction

Data for powder X-ray diffraction (XRD) were gathered using a Philips PW 1710 diffractometer with Cu K radiation (λ = 1.5418 nm) and Graphite monochromator at 45 kV, 30 mA, and 25 °C.

#### 2.6.4. Raman Spectroscopy

Spectroscopy data were collected using a WITec Confocal Raman Microscopy model Alpha 300R+y (Ulm, Germany). A frequency-doubled laser with an excitation wavelength of 532 nm and an output power of 38 mW was utilized in a typical experiment. Spectra were captured using a 50× Zeiss, EC Epiplan-Neofluar Dic objective (Oberkochen, Germany) with a numeric aperture of 0.8. Signals were detected in the range of 1024 × 127 pixel. The accumulation count was 30, and the integration time per pixel was 0.2 s. WITec Control program (Ulm, Germany) was used for data collection

#### 2.6.5. Magnetic Characterization

Magnetic moment measurements were performed with a commercial magnetometer based on superconducting quantum interference (Quantum Design, model MPMS-XL). A plastic straw was used as a sample holder, to which gelatin capsules filled with the powder samples were attached (126.3 mg in the case of sample Gd1 and 86.1 mg in the case of sample Gd10). The magnetic moment was measured as a function of temperature (down to 5 K) under a 100 Oe magnetic field, following both zero-field cooling (ZFC) and field-cooling (FC) processes. The reciprocating sample option (RSO) was used, providing a resolution up to the 10^−8^ emu range.

## 3. Discussion

### 3.1. Computational Fluid Dynamics (CFD)

First, with the aim of knowing the flow behaviour of the microfluidic systems, computational fluid dynamics (CFD) were performed using the software COMSOL. As explained above, one of the conditions that is essential to meet is that the flow behaves in such a way that can ensure a homogenous dispersion amongst the elements or phases involved in the process can be ensured.

In this line, looking at the results of the fluid model (Figure 1a) it can be confirmed that it is a continuous, homogeneous, linear flow with a direction normal to the section. With greater magnitude in the center of the section which decreases as it moves away from it radially. Studies were made for an input velocity of 3 mL h^−1^, so it can be safely affirmed that at this value of flow rate, a laminar, homogeneous, and constant behaviour will be acquired. In reference to the channel pressures, (Figure 1b), the response is such that it is irrelevantly higher at the entrances of stocks A_i_ and B with respect to the exit zone, ΔP ≈ 1 Pa. Herewith, it can be concluded that there are no large pressure changes throughout the system that could influence the correct mixture of the components. It is worth mentioning that the transition areas between the chip and the cables were the most critical zones due to the sudden change in the direction and geometry of the system and that is the reason why it was decided to carry out the study in 3D and not in 2D despite losing some precision in the results. These findings demonstrate laminar flow in all geometries and velocities, which is consistent with the Reynolds values (Appendix A). Even though the features may be anticipated in very basic circumstances, it is still quite simple to alter this profile by changing the mixing’s speeds and effectiveness to suit our demands. In this instance, however, only individual molecular diffusion via the interface between the fluid streams can result in the mixing of molecules in separate fluxes. Based on previous studies [57], it is known that nanoparticles with the correct size and size distribution may be achieved only if the growth ends at the proper moment, for example, owing to reagent depletion, while the particles are still in the acceptable nanometre size range, and particle aggregation is absent. In the case of turbulent flow, mass transport and mixing through eddies would generate instabilities in the system, which could cause uncontrolled particle nucleation and aggregation, therefore resulting in an inferior control of the process. Precisely, in order to achieve a narrow particle size distribution, the reaction should be regulated to ensure that nucleation occurs as quickly as possible and that no additional nucleation and coagulation of individual particles occurs throughout nanoparticle development. As mentioned, the homogeneous reaction and particle nucleation are diffusion restricted and occur only at the boundary of the two laminar streams, whilst a slower activation process occurs on a longer time scale, clearly demarcating the nucleation and growth stages, allowing for greater control of particle size and size distribution. On the other hand, there is a certain probability that the components could nucleate in the final step and generate solid particles during their passage through the chip. For this reason, a simulation of the final part of the system was carried out (i.e., the last channel of the chip together with the final tube) to verify that the introduction of NPs with analogous size and density values as the HAp nanorods does not influence the laminar flow and its distribution throughout the channel.

As it has been previously demonstrated [42], there is a relationship between the speed injected into the system and the size of the NPs obtained, in such a way that the higher the speed, the smaller the size.

To carry out this study, on the one hand, 15 NPs of 0.2 μm radius are computationally generated without initial velocity in a flow of 9 mL h^−1^ (conservative criterion) and their trajectory due to the drag force is traced with respect to time. On the other, static NPs are imposed within a flow under that speed and the possible temporary turbulence that the flow may suffer when interacting with them is checked.

Figure 2a,d shows both the initial and the final instant of the trajectory described by the NPs under the described geometry and the imposed conditions. As can be seen, both at the initial and at the final instant, the NPs show equivalent positions and distances with respect to the others and to the tube, that is, they describe equidistant trajectories despite the fact that each NP moves at a different speed. This fact exposes the fact that the NPs only suffer drag forces parallel to the flow and none other that generates possible turbulence. With the second simulation, Figure 2e,f we intended to check if the formation of solid elements in the channel generates a modification in the flow. For this purpose, in a pipe with a diameter of 0.7 mm and 25 mm in length, a flow with a rate of 9 mL h^−1^ and 125 spherical elements of radius 50 μm each are simulated, separated in planes of 25 elements. The chosen gap between planes was 3 mm and the separation between the elements in each plane was 0.2 × 0.2 mm with respect to their peers. The reason for this geometry and dimensions (microns) is the possibility of aggregation of several NPs in the pipe and consequent generation of elements in the micro dimension since the nucleation of individual NPs has a null effect on the flow due to its dimension in the nanoscale. As can be seen in Figure 2f, the only variation of the flow in the planes is due to the presence of the elements themselves, but when passing through each one of them, this flow becomes constant and homogeneous again.

In view of the results, it can be concluded that the system and conditions imposed are suitable for the synthesis of NPs and that the possible inconvenience of the compounds nucleating on the chip will not harm the final result.

### 3.2. Surface Characterization

The FE-SEM pictures obtained are represented in Figure 3. From a qualitative perspective, it can be seen in the untreated images that, in certain surfaces, the incorporation of gadolinium causes softness in the appearance of protuberances, that is, it reduces the rugosity with respect to those of pure HAp. That same behaviour has been already demonstrated in similar works, where the morphology of the samples changed considerably from long rod-like to short rod or mixed structures through doping [58,59]. As can be seen in the images, the crystallinity and homogeneity of the materials decreased as the gadolinium level increased, agglomeration was more common, and the form eventually changed to a mixture of needle-like geometries and irregular ellipsoids. There might be two explanations for the shift in crystal growth direction and morphology. On the one hand, the introduction of foreign ions changes the crystal surface energy. As explained, pure HAp was doped with a mixture of gadolinium chloride anhydrous with the presence of a phase of gadolinium oxychloride. Due to the fact that Gd^2+^ and GdO^−^ are cations and anions, respectively, they would be adsorbed to the different crystallographic planes of hydroxyapatite, affecting the free enthalpy on different planes in the HAp crystal. As a result, the growth rate of distinct crystal faces varies, and the aspect ratio decreases. On the other hand, the variation in morphology might be due to a shift in the crystal growth direction. Ca combines with six P–O tetrahedrons during HAp formation to produce a more stable Ca–P_6_–O_24_ polyhedron, which is expected to serve as a HAp growth unit on the c-axis [60]. With increased doped ion concentrations, more Gd^3+^ ions enter and substitute for Ca^2+^ in the HAp crystal lattice, allowing more anions in the surrounding environment to be absorbed. The adsorbed ions protect the contact where the crystal preferentially develops as a result. The presence of GdO^−^ also may prevent the development of the Ca–P_6_–O_24_ coordination polyhedron, preventing preferred crystal growth along the c-axis [59].

Quantitatively, to complete the analysis, image treatments and the calculations of the fractal dimension in the studied systems are shown in Table 1. The calculations were made for 9 images, 3 of each sample (pure HAp, Gd1, and Gd10) with an identical scale for greater precision, in this case, 200 nm for every 34 pixels. Each image was given the name I_i_ where i = 1, 2, 3 refer to those of pure HAp, i = 4, 5, 6 denote those of Gd1, and i = 7, 8, 9 allude to the Gd10 sample. As for Digitization 1, it was a process through ImageJ to obtain the roughness parameters through the grayscale of each image. In reference to Digitization 2, smoothing was performed on the previous digitizations to evaluate the z-dimension of the surfaces. Therefore, these digitizations correspond to the ZX plane. In them, each z-value was reduced by 0.5 to improve visualization. In the flat figures ZX, it can be perceived how there is a variation on a nanometric scale since in all cases there is a maximum variation of 1000 nm, but many elevations of lower height can be spotted, (Appendix A). Based on these results, R_q_ and R_a_ are lower in the presence of gadolinium and even lower if this element is in a higher concentration. This may be the mathematical explanation for the lower roughness seen by the naked eye. Likewise, said parameters in the samples with the presence of Gd vary considerably from one image to another on the same sample, which demonstrates heterogeneity even between images. These results contrast with the pure HAp values, where good homogeneity and closer parameters can be observed.

Regarding the values of *R_sk_* and *R_ku_*, they are very similar between all images, both in the same sample and between the different materials. In the case of *R_sk_*, all the values move between ±0.1 and, as explained, the disposition of the samples is to have a greater separation between peaks and less between valleys in all cases. A similar conclusion can be drawn from the evaluation of *R_ku_*. The behaviour of this parameter is analogous to the previous one and its variation is in the same order. It has a similar value in all the images regardless of whether they are of the same element. Thus, the roughness pattern is such that there are few deep valleys and high peaks and most have gentle ridges and troughs. Regarding the fractal dimension, all the samples revealed comparable behaviour, therefore, it can be quite safe to assume that all the samples have a comparable cross-linking lattice, both in geometry and in order.

### 3.3. Quantitative Elemental Analysis

In Table 2 and Table 3, data from component analysis of the Gd1 and Gd10 samples that were exposed to the EDX technique are presented. From them, two spectra are obtained that give information about the chemical composition of the samples (Appendix A). Spectra 1 and 2 correspond to sample Gd1 and spectra 4 and 5 to Gd10 respectively (Appendix A). The results demonstrate the presence of C, O, P, and Ca, HAp-forming elements. Similarly, on the one hand, the presence of N impurities is also detected, showing residues of the nitrite added in the synthesis. Likewise, it can be noticed how the Gd1 sample is more contaminated with elements foreign to the process followed, although in very low concentration. Regarding the presence of Gd, this is manifested in both samples and to a greater extent in Gd10 with respect to Gd1, as expected, although in a very heterogeneous way. Regarding the manifestation of Na and Cl, it should be attributed to the residues generated in the synthesis process of the first stage or union of the Ai and B stocks (first microchip).

These results evidence what was already concluded in the previous section about the heterogeneity of the samples with gadolinium, since in Gd1 there is an appreciable difference in Gd concentration depending on the area treated. On the other hand, in the Gd10 sample results probe that the degree of substitution between Gd and Ca is logically and expectedly higher while there is less deviation and greater homogeneity.

Regarding the structural characterization of the results using X-rays, a signal smoothing process was applied to reduce noise and after obtaining each spectrum, they were compared with the characteristic peaks of the reference compound provided, in this case, by the X’pert Highscore software database. The first study carried was aimed to determine if the hydrolysis reaction of GdCl_3_ to obtain gadolinium oxychloride was satisfactory (Equation (8)). The results of a sample collected from the reaction are detailed in Figure 4a. The spectrum acquired from the product of the reaction was compared with that of GdOCl and, as can be seen, the peaks obtained clearly correspond to that of the reference.

Similarly, Figure 4b shows the results of the different samples (pure Hap, Hap doped with Gd 1:1, and HAp doped with Gd 1:10) comparing them with the bibliographic reference of hydroxyapatite. It can be seen how the peaks of all samples correspond mainly to those of pure HAp, as in the case of (2θ) = 28.75°, 32°, 32.5°, 48°, and 53.5°, corresponding to (002), (211), (122), (213), and (411) reflections of HAp, respectively, with no evidence of the presence of other calcium phosphate phase. With this, it can be affirmed that all the samples present crystalline elements with peaks that could be indexed to the hexagonal HAp. Additionally, it was demonstrated that the Mg^3+^ or Gd^3+^ doping did not affect the HAp’s crystal structure. The result could be attributed to the relatively low amount of Mg^3+^ or Gd^3+^ in the samples, which was insufficient to affect the HAp’s crystal lattice. On the other hand, in the samples with the presence of gadolinium, the appearance of a new peak derived from the presence of this element can be seen at 2θ = 32.547° (lattice parameters: a = 3.614 Å, b = 3.614 Å, c = 5770 Å and indices (h k l) = (1 0 1)) with greater intensity in the sample with a higher concentration of gadolinium. This might be attributed to the entry of the Gd^3+^ ions on the lattice, because the hydroxyapatite structure is open and can host cations at the Ca^2+^ sites or interstice positions [61]. This result is shared with the bibliography [43]. In this way, the presence of this type of crystallization in the material is observed and the successful doping is confirmed.

### 3.4. IR Spectral Analysis

Figure 4c shows the results for the FT-IR spectra of the hydroxyapatite samples doped with gadolinium and magnesium. All the samples exhibit transmittance bands related to the characteristic vibrational modes of HAp [62,63], demonstrating that HAp was successfully synthesized. The weak shoulder at 961 cm^−1^ and the strong broadband at 1036 cm^−1^ correspond to ν3 and ν1 
${\mathrm{PO}}_{4}^{3-}$
 stretch vibrational modes, respectively, whereas the shoulder at 473 cm^−1^ and the strong bands at 565 and 604 cm−1 indicate the presence of the 
${\mathrm{PO}}_{4}^{3-}$
 group’s ν2 and ν4 bending vibrational modes. The bands associated with the stretch of the OH group can be found at 3569 and 631 cm^−1^. The wide band found from 3237 to 3663 cm^−1^ might be attributable to water vibrational modes. From its part, the bands appearing in the range of 1600–1400 cm^−1^ correspond to C═O, stretching, and in the range of 2400–2000 cm^−1^ relate to C═O═C, stretching, both being indicative of 
${\mathrm{CO}}_{3}^{2-}$
 substitution. The phosphate group (type B) or the hydroxyl group (type A) in the HAp lattice can both be replaced by carbonate ions [64]. Solutions in the temperature range of 50–100 ºC can precipitate B-type carbonated HAp [65]. In this case, a mixture of AB substitution types is formed (A type: 
${\mathrm{CO}}_{3}^{2-}$
 → OH^–^; B type: 
${\mathrm{CO}}_{3}^{2-}$
 → 
${\mathrm{PO}}_{4}^{3-}$
) [66] as the synthesis was done at room temperature and the hydrothermal treatment was done at temperatures well over 100 °C. The 
${\mathrm{CO}}_{3}^{2-}$
 ions substitution may have happened because the samples were exposed to ambient CO_2_ before to the hydrothermal treatment and, also, the disintegration of CTAB after hydrothermal treatment should be considered as another important factor affecting the process [25].

In the present case, the reactant used to dope the hydroxyapatite, GdCl_3_, exhibits an intense transmittance band at 1402 cm^−1^ attributed to the Gd–Cl interactions [25]. This band can be lightly shifted as a result of chemical interactions between Gd^3+^ and 
${\mathrm{PO}}_{4}^{3-}$
 [67]. So, and reinforcing the XRD measurements, gadolinium ions substitute calcium in the lattice, resulting in strong interactions with phosphate and preventing carbonate formation. In the case of this sample, the carbonate production in the Gd-doped Hap nanorods is inhibited. This phenomenon might be explained by an interaction between Gd^3+^ - 
${\mathrm{PO}}_{4}^{3-}$
 and the HA-Gd crystal, which may have resulted in a stronger bond between these two ions, making it harder for carbonate ions to permeate into the nanorods.

Furthermore, complementary information can be obtained from high-resolution confocal Raman microscopy. The spectrum of HAp:Gd1 from 0 cm^−1^ to 4000 cm^−1^ can be seen in Figure 4d, being (i) and (ii) the spectra corresponding to the two main bands detected by the WITec software. Blue areas can be associated with the Gadolinium dominant effect [68] and the bands present between 1026 cm^−1^ (*v*3), and 1073 cm^−1^ (*v*3) were assigned to the asymmetric *v*3 (P–O) stretching. The OH^−^ vibrational bands expected in the region of 630 cm^−1^ are not clearly identified, which is in good accordance with previous studies with doped Hydroxyapatite [69]. From the Raman mapping, it can be clearly seen that the Gadolinium has effectively been introduced into the HAp matrix and that its distribution is consistent and homogeneous, being present both in the valleys and peaks of the studied sample. As expected, these results are in accordance with the previous conclusions drawn from the surface characterization and the quantitative elemental analysis where it has previously demonstrated that the Gd-doping has a perceptible but limited impact in the surface morphology of the final material.

### 3.5. Magnetic Field

Materials with magnetic ions in their composition present a magnetic susceptibility that is given by the Curie law, χ = C/T, where T is the temperature and C (the Curie constant) depends on the type and concentration of the magnetic ions. This expression, which characterizes the paramagnetic materials, results from the interaction of the ions with an applied magnetic field H (which tends to align the ions’ magnetic moment in its direction), and from the disorder due to thermal agitation. When the exchange interaction between the magnetic ions is not negligible, they can self-align below the so-called Curie temperature, T_C_. Materials with this property, known as ferromagnetism, are strongly magnetized below T_C_, even in the absence of an applied magnetic field. Above T_C_, the magnetic susceptibility of these materials is given by the Curie-Weiss law, χ = C/(T − T_C_). Nanoparticles of ferromagnetic materials can behave as paramagnetic even below T_C_, if the thermal agitation energy is enough to collectively flip their magnetic moments, a phenomenon known as superparamagnetism. Superparamagnetic materials present high magnetic susceptibility, orders of magnitude larger than that of paramagnetic materials, and their magnetic moments can be completely aligned under accessible applied magnetic fields. Another class of magnetic materials is antiferromagnets, for which the exchange interaction tends to anti-align the magnetic ions below the so-called Néel temperature T_N_, and at higher temperatures their magnetic susceptibility is given by χ = C/(T + T_N_). Finally, all materials present a negative (diamagnetic) temperature-independent contribution to the magnetic susceptibility coming from orbital electrons, which is overcome by the aforementioned magnetic contributions, if present.

The temperature dependence of the magnetic moment (m) of samples Gd1 and Gd10 is presented in Figure 5a,b. Typical paramagnetic behaviour is observed in both samples, with no appreciable differences between the ZFC and FC modes. The bulk-Gd ferro-paramagnetic transition is not present near 293 K [70], suggesting that clusterization of Gd ions is not taking place. A m(H) cycle up to 7 T was performed at 310 K. As can be seen in Figure 5c,d, a linear paramagnetic response is observed in both samples, with no appreciable hysteresis.

The solid lines in Figure 5a–d are fits to a Curie-Weiss function:
(13)
$$m=m_{0}+\frac{C}{T-\theta }H,$$

where *m*_0_ is a temperature-independent contribution of diamagnetic materials and the sample holder, *θ* is a temperature parameter (if *θ* > 0 it corresponds to the Curie temperature *T_C_* of a ferromagnetic material, and if *θ* < 0 it corresponds to minus the Néel temperature *T_N_* of an antiferromagnetic), *T* is the temperature, *H* is the applied magnetic field, *C*

$\equiv {\mathrm{Np}}^{2}\mu_{B}^{2}\mu_{0}/3k_{B}$
, *N* is the number of Gd ions, *p* = 7.98 [70] is the effective Bohr magneton number of Gd ions, 
$\mu_{B}$
 the Bohr magneton, 
$\mu_{0}$
 the vacuum magnetic permeability, and *k_B_* the Boltzmann constant. In both samples a slightly negative *θ* value was obtained (−0.89 ± 0.02 K for sample Gd10 and −2.07 ± 0.04 K for sample Gd1). A similar result has been observed in rare-earth oxide nanoparticles [71], and it suggests the emergence of an antiferromagnetic phase below the Néel temperature *T_N_* = −*θ*. The fits also lead to *N* = (10.52 ± 0.03) × 10^19^ and (5.75 ± 0.02) × 10^19^ for samples Gd10 and Gd1, respectively. This corresponds to a Gd mass of 27.5 mg in sample Gd10 (~22% of the total), and 15.0 mg in sample Gd1 (~17% of the total). These results are in reasonably good agreement with the outcomes of the EDX analysis, as the mass of Gadolinium corresponds to 20–30% of the total mass in the studied regions.

## 4. Conclusions

After the synthesis, computational studies, and characterization techniques were carried out, it can be established that an innovative route has been obtained to obtain HAp elements on the nanometric scale with magnetic properties. Additionally, it should be highlighted the coherence and correlation shown between the different procedures gave quite similar results and conclusions. Although it is true that there are discrepancies in the surface morphology of the doped samples with respect to the negative control, the considerable heterogeneity in the presence of certain elements shown by the EDX technique, and the non-linearity in the presence of Gd with respect to the initial concentrations added. It can be seen that the X-ray results have been satisfactory, nanometric surfaces and patterns similar to the negative control were obtained, and gadolinium is present in the doped samples, upgrading the bio-ceramic nanoparticles with paramagnetic and theragnostic properties.

## Figures and Tables

**Figure 1 nanomaterials-13-00501-f001:**
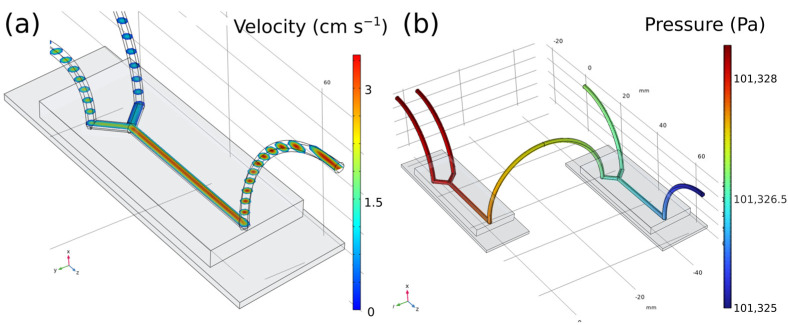
Computational fluid dynamics (CFD) from a simple chip (**a**) and the system of two connected in series (**b**). In the first image a) the flow velocity inside the channels of the second microchip can be seen. As expected, a greater magnitude is found in the center of the section, decreasing as it moves away from it, radially. The second image (**b**) depicts the pressure changes, and it can be concluded that there are no large pressure changes throughout the system that could influence the correct mixture of the components.

**Figure 2 nanomaterials-13-00501-f002:**
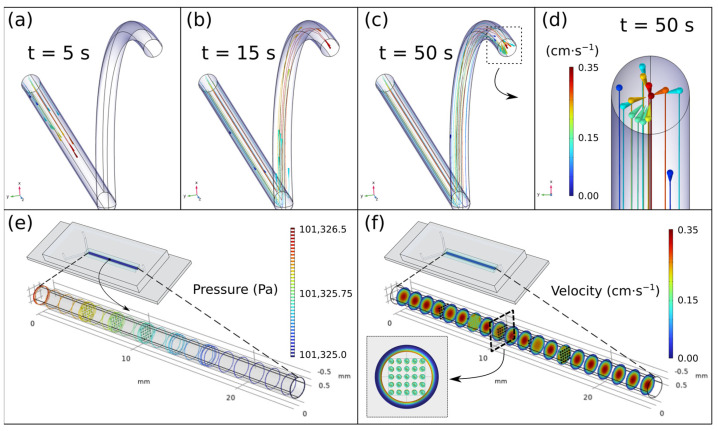
From (**a**–**c**). Particle position at different time points. Each particle describes equidistant trajectories. (**d**) Close-up of the end of the output tube. (**e**,**f**) Pressure and velocity simulations of the inlet channel assuming a grid of 25 elements on each panel. Results confirm that the presence of a small solid dragged by the flow does not affect the homogeneity.

**Figure 3 nanomaterials-13-00501-f003:**
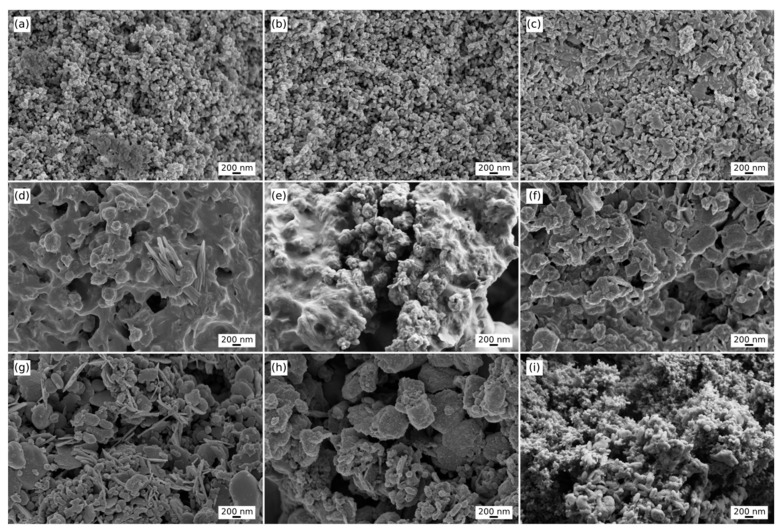
SEM images of the different samples. Images from the upper row (**a**–**c**) depict the surface of pure HAp nanorods. The central row (**d**–**f**) corresponds to the HAp:GD1 samples, and the lower row (**g**–**i**) represents the results for the HAp:GD10 material. It can be seen in the untreated images that the incorporation of gadolinium causes softness in the appearance of protuberances and a general change in the overall surface.

**Figure 4 nanomaterials-13-00501-f004:**
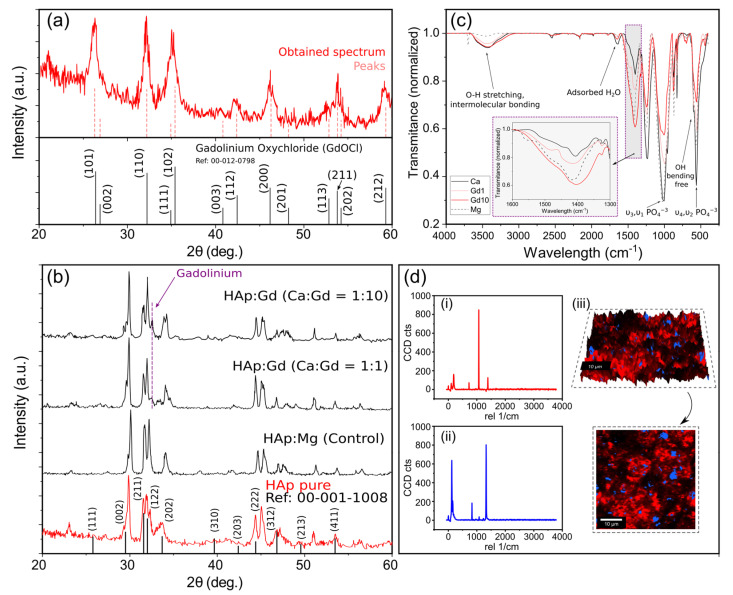
(**a**) X-ray diffraction obtained for the resultant material from the hydrolysis reaction of GdCl_3_. Major peaks correspond to Gadolinium Oxychloride (ICDD Ref: 00-012-0798), corroborating the chemical reactions predicted by the bibliography. (**b**) FTIR spectra obtained for the four studied samples. In the inset, the critical range of 1600–1400 is zoomed in. (**c**) X-ray diffraction curves for the different samples. As expected, the Sample with a higher concentration of Gd in the initial step of the synthesis, produces a higher peak at 2θ = 32.547° (lattice parameters: a = 3.614 Å, b = 3.614 Å, c = 5770 Å and indices (h k l) = (1 0 1)), Probing a successful substitution of Gd in the Ca sites. (**d**) Raman spectra in the 0–4000 cm^−1^ wavenumber range of HAp:Gd1. Spectra (i) and (ii) represent the two main bands detected by the software. Figure (iii) shows the optical images of the compound distribution.

**Figure 5 nanomaterials-13-00501-f005:**
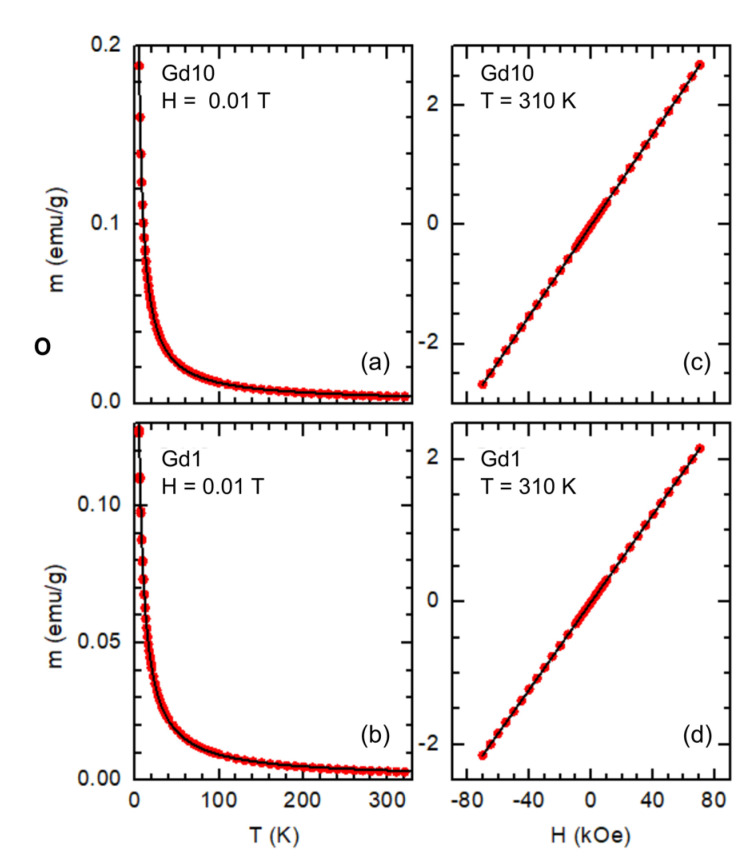
Temperature (**a**,**b**) and magnetic field (**c**,**d**) dependences of the magnetic moment of samples Gd10 (126.3 mg) and Gd1 (86.1 mg). The lines are fits of a Curie-Weiss function, Equation (13). Demagnetization effects are negligible (the correction would be ~2–3% at 5 K and strongly decreases on increasing the temperature) and were not taken into account.

**Table 1 nanomaterials-13-00501-t001:** Roughness and diffractal coefficients of FE-SEM images.

Image	$R_{q}$ [nm]	$R_{a}$ [nm]	$R_{sk}$ [nm]	$R_{ku}$ [nm]	$D_{f}$
$I_{1}$	122.342	114.912	1.152	1.411	1.802
$I_{2}$	114.355	105.54	1.189	1.512	1.792
$I_{3}$	112.207	104.362	1.179	1.492	1.821
${\overset{¯}{I}}_{123}$	116.301	108.271	1.173	1.472	1.805
$I_{4}$	96.05	92.06	1.118	1.341	1.886
$I_{5}$	133.001	118.683	1.267	1.733	1.804
$I_{6}$	93.641	86.576	1.203	1.591	1.834
${\overset{¯}{I}}_{456}$	107.564	99.106	1.196	1.555	1.835
$I_{7}$	84.912	78.723	1.177	1.48	1.723
$I_{8}$	79.484	72.189	1.225	1.624	1.735
$I_{9}$	108.081	91.679	1.203	1.773	1.86
${\overset{¯}{I}}_{789}$	90.826	80.864	1.202	1.626	1.773

**Table 2 nanomaterials-13-00501-t002:** EDX values obtained for each element present in the samples.

Element	Esp. 1	Esp. 2	Esp. 4	Esp. 5
C	36.60	14.84	28.15	20.84
N	-	6.59	11.05	5.84
O	6.59	29.56	28.93	28.17
Na	20.53	14.57	11.62	12.91
P	0.39	4.69	1.64	4.03
Cl	30.47	1.63	0.31	0.89
Ca	0.44	3.22	0.02	0.02
Other	2.69	0.22	-	-
Gd	2.30	24.67	18.29	27.3
Total	100	100	100	100

**Table 3 nanomaterials-13-00501-t003:** Statistical treatment for quantitative elemental analysis.

Element	C	N	O	Na	Al	P	Cl	Ca	Other	Gd
Max. Gd1	36.60	6.59	29.56	20.53	0.22	4.69	30.47	3.22	2.69	24.67
Min. Gd1	14.84	6.59	6.59	14.57	0.22	0.39	1.63	0.44	2.69	2.30
Mean Gd1	25.72	-	18.08	17.55	-	2.54	16.05	1.83	-	13.48
Typical deviation	15.38	-	16.25	4.21	-	3.04	20.39	1.97	-	15.82
Max. Gd10	28.15	11.05	28.93	12.91	-	4.03	0.89	0.02	-	27.3
Min. Gd10	20.84	5.84	28.17	11.62	-	1.64	0.31	0.02	-	18.29
Mean Gd10	24.5	8.44	28.55	12.26	-	2.84	0.6	-	-	22.79
Typical deviation	5.17	3.68	0.54	0.91	-	1.69	0.41	-	-	6.37

## Data Availability

The data that support the findings of this study are available from the corresponding author, R.R., upon reasonable request.

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
