# Peer review of "Microfluidic Fabrication of Gadolinium-Doped Hydroxyapatite for Theragnostic Applications"

_nanomaterials, 2023, doi:10.3390/nano13030501_

Round 1

Reviewer 1 Report

1. The state of the fluid in the chip is closely related to the Reynolds number, generally less than 1000 is laminar flow, and above 4000 is turbulent flow. The author lists the calculation formula in experiment 2.4 Computational parameterization formula, but the calculation process and results are not seen in the discussion section.

2. In line 324, the author proposes that the internal fluid of the Y-chip is laminar flow, and laminar flow generally leads to uneven particle size and large particle size. (reference, Microfluidic Methods for Fabrication and Engineering of Nanoparticle Drug Delivery Systems. ACS Applied Bio Materials 2020, 3, 107-120. )

3. The internal pressure of the two Y-shaped chips is relatively high, how does the author ensure the sealing of the interface. Why not consider integrating on a chip

Reviewer 2 Report

nanomaterials-2183647-peer-review-v1

The authors provided an interesting way to manufacture Gd-doped Hydroxyapatite 2 for theranostic applications. The design of the experiment is good and extensive characterizations were carried out to assess the proposed method. Despite the large standard deviations observed by EDX, the results seem promising for trials made in a laboratory environment (i.e. without process control and qualified mfg methods…). This research is of certain interest to the readers but some parts of the discussion should be improved (see the detailed list of comments hereafter), notably the discussion of the magnetic test results. Also, to better show the potential of this method, the measured variability induced by the process and the devices should be better compared to state-of-the-art manufacturing methods.

Comments/recommendations/questions:

The authors mentioned that laminar flows are required to get homogeneous diffusion between the phases, it could be great to elaborate further and explain why a turbulent flow, which also generates mass transport and mixing through eddies in addition to diffusion, is of lesser interest here.

Line 297: what is exactly the shape of the sample? According to the state-of-the-art about magnetic measurements of such Gd compounds, did you check that the demagnetization factor does not matter here?

Line 577, please double-check the notation for the Curie temperature

Please also define theta (Weiss constant?)

Line 579: Weiss constants are very small, please indicate if the sign of this constant is significant or not according to measurement noise and R2 fit parameter. Did you expect such small values? Is it supported by previous research?

I understand that it is very convenient to use emu system, please double-check if it is acceptable as units in the journal – units can also be specified in lines 577-578.

It is a bit confusing to use m3 with emu (cm is usually used instead – cgs system), maybe SI units all along the discussion may be preferred.

Line 582: what is the most relevant? volume or molar reference for C?

Line 583: unit for C? emu/cm3?

Could be helpful to elaborate more on the magnetism of Gd3+ ion (spin state, nb of Bohr magneton, or the magnetic moment per ion…), together with indications about pure Gd crystals (Curie temp…) to double-check that there is no aggregation of ferromagnetic clusters.

Round 2

Reviewer 2 Report

I would like to thank the authors for conscientiously addressing all the comments raised in the first revision of the manuscript. The revised version of the manuscript can be published in its current form in Nanomaterials.